# Vaccines against Emerging and Neglected Infectious Diseases: An Overview

**DOI:** 10.3390/vaccines10091385

**Published:** 2022-08-25

**Authors:** Larissa Vuitika, Wasim A. Prates-Syed, Jaqueline Dinis Queiros Silva, Karin P. Crema, Nelson Côrtes, Aline Lira, Julia Beatriz Menuci Lima, Niels Olsen Saraiva Camara, Lena F. Schimke, Otavio Cabral-Marques, Mohammad Sadraeian, Lorena C. S. Chaves, Gustavo Cabral-Miranda

**Affiliations:** 1Department of Immunology, Institute of Biomedical Sciences IV, University of São Paulo (ICB-IV/USP), São Paulo 05508-000, SP, Brazil; 2Institute of Research and Education in Child Health (PENSI), São Paulo 01228-200, SP, Brazil; 3Laboratory of Research in Infectious Diseases (LAPI), Hospital Prof. Edgard Santos, Federal University of Bahia, Salvador 40110-060, BA, Brazil; 4Institute for Biomedical Materials and Devices (IBMD), Faculty of Science, University of Technology, Sydney, NSW 2007, Australia; 5Department of Microbiology and Immunology, School of Medicine, Emory University, Claudia Nance Rollins Building, Atlanta, GA 30329, USA

**Keywords:** neglected diseases, emerging infectious diseases, vaccines

## Abstract

Neglected Tropical Diseases (NTDs) are a group of diseases that are highly prevalent in tropical and subtropical regions, and closely associated with poverty and marginalized populations. Infectious diseases affect over 1.6 billion people annually, and vaccines are the best prophylactic tool against them. Along with NTDs, emerging and reemerging infectious diseases also threaten global public health, as they can unpredictably result in pandemics. The recent advances in vaccinology allowed the development and licensing of new vaccine platforms that can target and prevent these diseases. In this work, we discuss the advances in vaccinology and some of the difficulties found in the vaccine development pipeline for selected NTDs and emerging and reemerging infectious diseases, including HIV, Dengue, Ebola, Chagas disease, malaria, leishmaniasis, zika, and chikungunya.

## 1. Introduction

Neglected Tropical diseases (NTDs) are a group of infectious and non-infectious diseases highly prevalent in tropical and subtropical regions, which affect over 1.6 billion people annually [1]. Most of these people are marginalized, afflicted by poverty, wars, and stigma, and do not have access to drinkable, clean water [2,3]. Currently, WHO recognizes 20 diseases, whose causative agents comprise viruses, bacteria, fungi, helminths, protozoa, and snakebite envenomation (Table 1) [1,2,3,4].

Fortunately, most of these diseases have decreased in prevalence in the past decade, and the most significant declines are attributed to Dracunculiasis (Guinea worm) (99%) and African trypanosomiasis (73%) [4]. In contrast, the prevalence of Dengue fever (75%) and cutaneous leishmaniasis (29%) have increased [4]. Therefore, it is necessary to go deeper and look over the successes and failures of these aforementioned cases to understand better what lies ahead and why investment in research is so significant [4].

Besides NTDs, Emerging Infectious Diseases (EIDs) are another group that cause a major burden in global health. However, whereas the NTDs are mostly ancient diseases, EIDs are caused by recently discovered pathogens of a zoonotic origin. This classification first obtained international attention in the 1960s with the emergence of viral hemorrhagic fevers, such as Crimean–Congo hemorrhagic fever and Ebola fever. In the 1980s, these diseases received greater attention with the emergence of other severe syndromes, especially HIV/AIDS [5].

Throughout the past few decades, these diseases have been emerging and re-emerging in different regions, some of which caused pandemics and others were controlled. Furthermore, the attention to EIDs with pandemic potential was reflected by the establishment of the WHO Emergency Committee in 2005 [6,7]. Since then, five Public Health Emergencies of International Concern were declared: the pandemic of HN1 influenza in 2009; the polio resurgence in 2014; the Ebola epidemic in West Africa in 2014; the emergence of Zika virus in the Americas in 2016 [7]; and now, the tragic epidemic caused by COVID-19 (Coronavirus disease-19).

Several factors can contribute to the emergence of pathogens, such as the complex interactions between infectious agents, hosts, and the environment, which are mostly unpredictable [8,9]. This is because the decrease in natural resources leads to the invasion of wild animals in urban and rural areas, which can be intermediate hosts or vectors of a new or re-emerging disease, some of which can also be NTDs, including Dengue, leishmaniasis, Chagas disease, and malaria [6,7].

Vaccination is one of the most efficient ways to prevent and control infectious diseases [10,11]. This is especially true for smallpox, the first and still the only eradicated human disease [12]. Besides smallpox, several other infectious diseases were expressively controlled with vaccination [13]. Although many technologies have been developed in the past few decades (Figure 1), the development of a vaccine that induces a safe and protective immune response is often a very challenging task, especially against most of the NTDs and EIDs [14]. In addition, it is imperative to make vaccination accessible to as many countries as possible. This can only be achieved if there are enough resources for the researchers to understand each target and the immune response underlying the natural infection, which is rare for NTDs. With this knowledge, it is possible to formulate highly scalable and cost-effective strategies using vaccines through pre-clinical and clinical trials (Figure 2).

The lack of vaccines against NTDs and EIDs is not surprising. There are several difficulties in the development of new vaccines, mainly the low investment in research, lack of industry interest, and the complex life cycle of some of the pathogens. Another problem is the recent rise of the antivaccination movement, which not only impacts public health and vaccination advances, but also social and political stability [1].

## 2. An Overview of NTDs and Vaccine

### 2.1. Ebola Virus (EBOV)

One of the most feared diseases of a new epidemic is the Ebola virus (EBOV) due to high mortality in infected humans (50% to 90%) [15]. In the African continent (2014–2016), a recent outbreak of EVD (Ebola Virus Disease) killed 11,000 people [16]. It is an infectious disease with severe clinical signs, such as hemorrhagic fever, diarrhea, severe fatigue, anorexia, abdominal pain, hiccups, myalgia, vomiting, confusion, and conjunctivitis that can lead to vision loss [15].

According to WHO, the two licensed vaccines are anti-EBOV [1]: rVSV-ZEBOV, a single-dose vaccine manufactured by Merck; and the two-dose Ad26.ZEBOV/MVA-BN-Filo, manufactured by Janssen. The rVSV-ZEBOV uses a recombinant vesicular stomatitis vector (rVSV), and the target antigen is the surface glycoprotein, which proved to be critical in inducing protective humoral immune response. During the recent outbreak in the Democratic Republic of the Congo (2018), rVSV-ZEBOV-GP was used for ring vaccination in the affected area [17]. The Ad26.ZEBOV/MVA-BN-Filo vaccine uses two no-replicant viral vectors and is applied in heterologous two doses. In the first dose, the recombinant non-replicant adenovirus type (Ad26) vector expressing Zaire Ebola virus glycoprotein is applied; the second dose corresponds to a non-replicate, recombinant, and modified vaccinia Ankara (MAV) encoding the glycoproteins from the Zaire Ebola virus, Sudan virus, Marburg virus, and nucleoprotein from the Tai Forest virus [18]. The two doses are applied with intervals of 28, 56, and 84 days between doses and elicits robust humoral and cellular responses that persist for one year after vaccination [18].

The Ebola vaccine is the first licensed for human use based on the viral vectors’ platform, which has provided key scientific information to generate some of the COVID-19 vaccines. With viral vector technology, it is possible to produce vaccines capable of inducing protective immunity by antibodies and T cell responses, in addition to showing safety, as they employ viruses that do not cause serious disease in humans [19,20]. However, the replicating viral vectors may limit their use in immunocompromised people, while the use of non-replicating viral vectors loses some of the natural ability to induce a full immune response, such as cellular immunity [19]. In addition, the previous immunity against the viral vectors can significantly reduce the effectiveness of the vaccine, as neutralizing antibodies can prevent the vectors from infecting the cells of immunized persons, limiting them to producing the antigens of vaccine interest [20]. Some strategies to overcome these barriers can be executed through heterologous vaccination, in addition to the correct choice of vectors that do not have previous immunity.

Other vaccines against Ebola were studied and are undergoing clinical trials in humans, including DNA vaccines and adenovirus-based vaccines (rVSV, rAd5, rAd26, and ChAd3/cAd3) [16]. In addition, several others are being studied in preclinical trials in some animal models (mice, pigs, and non-human primates) as inactivated vaccines, DNA vaccines, VLPs, Virus-Like Replicon Particles (vRPs), and recombinant viral vectors [7,8].

### 2.2. Dengue Virus (DENV)

The development of a vaccine against DENV is challenging, due to the four antigenically distinct serotypes that can trigger an adverse effect. In 2015, the first Dengue vaccine was approved (CYD-TDV “Dengvaxia”) and recommended in several countries, especially those that are highly endemic for DENV [10]. This vaccine is tetravalent, attenuated, and has a 17D reserve of the yellow fever virus. However, the Dengvaxia is limited to use in people who are over nine years old and seropositive for Dengue. This limitation gave voice to many scientific discussions. However, the fact is that the initial studies with this vaccine showed acceptable efficacy and safety, and the risk induced by it was not observed during Clinical Trial Phase III. This was not sustained after further analyses, that subsequently identified risks associated with the Dengvaxia vaccination programs, especially with the risk of ADE (antibody-dependent enhancement) in children under nine years old, who had no previous DENV infection. Indeed, several studies were carried out and, although there is still a contradiction in this regard, the fact that vaccination generates a significant risk for a certain group, such as those under nine years of age, led to this restriction [21,22,23,24,25].

Two other vaccines are in phase III of clinical trials, and both use the attenuated form of the Dengue virus. One of them, the DENVax, consists of attenuated DENV-2 and contains the prM and envelope proteins of DENV-1, DENV-3, and DENV-4. The difference between DENVax and Dengvaxia is the presence of non-structural proteins (NS), due to the DENV-2 backbone that favors the generation of cellular responses mediated by T cells, and the production of antibodies against NSI that are associated with cross-protection for humoral responses. The results of phases I and II show the production of neutralizers and cellular responses related to T cells [24]. Another tetravalent vaccine is being developed by the U.S. National Institutes of Health (NIH). This vaccine is currently in phase III of clinical trials in Brazil, which is composed of three attenuated virus serotypes, and the fourth component is a chimeric virus in which the DNV-2 prM and the E proteins replace DENV-4. This vaccine performed well and was safe in phase I and II trials [24,25].

Although the development of these vaccines against DENV has presented important scientific advances, there is a need to consider that the use of cross-reactive antigens among the serotypes raises a concern about the possibility of inducing Dengue hemorrhagic fever, as it happens naturally and with the Dengvaxia [22]. Thus, the use of attenuated virus technology will always raise uncertainty about vaccine safety. Therefore, the use of other platforms, such as VLPs, subunit vaccines and/or new genetic technologies, can solve this problem. Moreover, the vaccine strategies against DENV should be improved to produce high titers of the specific neutralizing antibodies equally against the four types, because it has not happened with the licensed vaccine and others that are currently in clinical studies. In addition, the vaccine must be able to generate a humoral and cellular immune response, as well as protection and memory, in all of the groups of the population, without causing Dengue fever.

### 2.3. Chikungunya Virus (CHIKV)

CHIKV is an arbovirus that was first isolated in Tanzania in 1952, with the first epidemic record in 1952–1953. There are three circulating genotypes and one serotype corresponding to the Chikungunya virus [26]. Nowadays, four lineages of CHIKV circulate in the world [26]: (1) East Central and Southern African (ECSA) lineage; (2) West Africa lineage; (3) Asian lineage; and (4) Indian Ocean lineage (IOL). The studies revealed no difference in virulence between the CHIKV strains representing the different lineages [27].

This virus has been able to cause epidemics in various parts of the world over the past few decades and it is estimated that more than 3.4 million people are infected globally. CHIKV infection has become a worldwide threat and caused serious medical problems, such as fever, muscle pain, headache, nausea, fatigue, and rash, and most of the patients exhibit joint pain for years after the onset [28,29]. Despite being a huge problem for public health, to date, there are no vaccines licensed for human use to prevent CHIKV. However, there are some vaccines in clinical trials [30]. 

The development of a safe and efficient vaccine anti-CHKV is fundamental to prevent new outbreaks. Moreover, the population of *A. aegypti* and *A. albopictus*, the mosquitoes that transmit this, among others, arbovirus, continues to grow due to pesticide resistance [31]. A vaccine developed using a VLP platform conjugated with the viral capsid E1 and E2 proteins is safe and efficient in producing high neutralizers against the different CHIKV genotypes [32]. Another three vaccines are in the phase of clinical studies using the attenuated virus (CHIKV/IRES), the measles vector platform (NCT02861586 and NCT03101111), and the chimpanzee adenovirus vector (ChAdOx1 Chik) [33]. The use of different vaccine platforms opens the possibility of having more than one vaccine, which can be offered to different population groups, in addition to opening the possibility of better distribution of it to different parts of the world, especially for those who suffer the most from this disease.

### 2.4. Zika Virus (ZIKV)

ZIKV is another arboviral disease that can cause severe comorbidities, such as microcephaly, brain death in newborns, and fetal death [34,35]. Although sporadic ZIKV outbreaks have occurred worldwide, it is imperative to be prepared for new outbreaks. Several vaccine strategies were studied, and many of them are in the preclinical phase and others in the clinical phase [36,37,38].

Unlike other flaviviruses that have more than one serotype, only one corresponding serotype for ZIKV was identified. This fact may favor the development of a single vaccine against this disease. However, some of the studies have shown that the antibodies against ZIKV may have a cross-reactivity against DENV and may lead to antibody-dependent enhancement (ADE) [39,40]. Therefore, it is necessary to be careful, because the two flaviviruses are transmitted by the same mosquito vector and they circulate in the same environment [41,42].

Different platforms have been developed and tested for the production of anti-ZIKV vaccines. These include recombinant protein antigen vaccines, inactivated whole virus vaccines, DNA and RNA vaccines, live attenuated vaccines, viral vector vaccines, and VLPs [43,44]. In addition, the development of peptide vaccines based on immunoinformatic approaches was proposed [45,46]. Subunit vaccines express two virion surface glycoproteins of prM and E, using DNA, mRNA, or viral vectors (e.g., adenovirus) [27]. The experimental studies demonstrate that these vaccines induce blocking antibodies and CD8^+^ T-CTL to the structural proteins, both desirable for protective immunity. Until now, there is no licensed vaccine, and many problems remain of the evaluation of effectiveness and possible adverse outcomes [47].

### 2.5. Other Neglected and Emerging Infectious Diseases without Vaccines

The above topics cite the progress of research and clinical trials in neglected and emerging diseases. However, other severe diseases affect humans, and vaccines are still needed for their control and dissemination.

HIV is the etiologic agent for AIDS. According to the WHO, there were an estimated 37.7 million people living with HIV at the end of 2020, over two-thirds of those were in the African Region [1]. Besides that, 680,000 people died from HIV-related causes, and 1.5 million people acquired HIV [1].

Fortunately, effective antiretroviral drugs exist for patients with HIV infections, and their application as pre-exposure prophylaxis (or PrEP) helps to prevent infection in at-risk people [48]. Despite the success of the antiretroviral therapy, most people worldwide, including the poorer people, do not have access to this treatment. Because of that, a vaccine is essential in this context for preventing the population in these risk areas.

In the last few years, since the discovery of HIV and the global epidemic, several vaccines were studied, some in clinical trials with prophylaxis and/or treatment of the people previously infected with the virus. The main antigenic target of the vaccine studies is the envelope protein (Env) gp120 or gp160, a glycoprotein target neutralizing the antibodies with the highest mutagenic rates that permit the escape of the virus from the immune system [49]. Another difficulty is the unsuitable animal model for the studies presenting the virus’s pathology.

The first efforts focused on stimulating CD8+ cells and producing neutralizing antibodies. Unfortunately, it did not achieve good results in some trials from 1980 to the early 2000s, such as Vax003 and Vax004, which used envelope proteins and alum as the adjuvants. Some of the viral vector vaccines with HIV-1 genes were developed to stimulate CD8+ cells but also aimed to induce the production of other viral proteins. Two trials, STEP and Phambili, ended before the predicted final date due to not increasing the immune defense, or decreasing the viral load in phase III [50].

The second effort was the prime-boost strategy to activate the B cells response through recombinant gp120 viral proteins; the RV144 trial achieved an efficacy of 60% in the first year after the administration, followed by 31% in 3.5 years [50]. The third and current effort relies on stimulating the production of polyvalent mosaic antigens by combining the strategies above. Vaccine engineering creating mosaic vaccines has been showing promising results since 2014, improving daily. It is important to highlight that, even with all these new technologies, effective protection against HIV is still challenging.

Chagas is another neglected and emerging infectious diseases without vaccines. It is a parasitic infection caused by *Trypanosoma* sp. and transmitted by triatomine insects. The causative agent was discovered in 1909 and yet it remains without a prophylactic, effective, and licensed vaccine. According to the WHO, it is estimated that 6–7 million people are affected by this disease [1]. The anti-parasitics are the only effective treatment in the initial phase of the disease, and the best prophylactic treatment is the vector control [1]. However, a vaccine combined with drug treatment would be of great importance for treating chronic patients. In addition, using vaccines as a prophylactic measure in endemic countries would help to contain the disease.

There has been a wide variety of vaccine platforms tested in animal models including live, attenuated vaccines, DNA, viral vectors, adenovirus, and subunit vaccines (recombinant peptides and proteins) [51,52]. The preclinical studies of therapeutic vaccines showed that some of the vaccines decrease the parasitic load (in the blood, cardiac, and skeletal muscle tissues), improve animal survival, and decrease the cardiac damage disorder. Finally, some of the platforms are already being tested in clinical trials at different stages of experimentation [51,52].

Following the same context, another disease for which a vaccine has not yet been engineered is Leishmaniasis, caused by the protozoan *Leishmaniose* sp. It is estimated that 1 million people become infected every year [1]. There are two types of clinical manifestations of Leishmaniasis: (1) zoonotic cutaneous Leishmaniasis (ZCL); and (2) anthroponotic visceral leishmaniasis (AVL), that are transmitted between animals, such as dogs, and humans, as well as from human to human, by the sand fly insects’ vector. Currently, the treatment involves chemotherapy, but it is highly toxic.

There are four licensed vaccines for veterinary use, but none for human use. We can mention some of the vaccines that have been tested in clinical trials: (1) a first-generation autoclaved Leishmania vaccine (LEISH-F1, precipitated by Alum and tested with BCG in a clinical trial Phase II) proved to be safe and immunogenic in 76% of human volunteers; (2) LEISH-F3 is composed of the *L. donovani* FML glycoprotein fraction and *L. infantum* 24-c-methyltransferase (SMT) and revealed a robust immune response against visceral Leishmaniasis, high levels of cytokine production in response to each vaccine component, and was considered a safe and well-tolerated vaccine in human volunteers. Other antigens and delivery platforms were developed, but many studies are still necessary to address the deficit in preventing Leishmaniasis.

Malaria disease is caused by *Plasmodium sp.* parasite. In 2019, it caused 435 million infections and 435 hundred deaths globally. Despite these impressive numbers of morbidity and mortality, malaria is not considered to be a neglected disease. Chemotherapy is the actual treatment, although it is often ineffective and toxic. The future gains against malaria depend on the availability of an effective vaccine, which is complicated by the complex life cycle of the parasites and other malaria strains [47].

Until then, the malaria strategy for developing vaccines is focused, according to the parasite life cycle, as pre-erythrocytic vaccines (PEV), blood-stages vaccines (BSV), and transmission-blocking vaccines (TBV) [53]. The PEV targets the antigens from sporozoites that clinically initiate human infection after a mosquito inoculates sporozoites into the skin. An example of PEV in phase IV clinical trials is the RTS,S AS01 vaccine. In 2019, a pilot implementation program was launched in Malawi, Ghana, and Kenya to assess the protective benefits and safety of RTS,S AS01. The previous clinical trials (including a large-scale phase III trial) in sub-Saharan Africa confirmed that the RTS,S malaria vaccine reduces malaria significantly, including life-threatening severe malaria in young children [53], which led to the WHO recommending the use of this vaccine worldwide for this group, in the regions that present a high prevalence of *P. falciparum* transmission. The BSV focuses on blocking the asexual parasite forms that undergo repeated multiplicative cycles in erythrocytes and cause disease and death. The TBV uses a surface antigen of the mosquito/ sexual stage of parasites (gametes and zygotes) to induce antibodies that kill the parasites in the mosquito blood meal and interrupt the parasite transmission through the vector. In model animals, the antigen fs25 was highlighted because it induced equal or greater serum transmission-blocking activity when compared with other antigens [53], and has been the focus of clinical trials published to date.

Because of the complex life cycle of *Plasmodium sp.* and its ability to circumvent the immune system, there are many difficulties to develop effective vaccines. New studies and approaches must be performed, as well as the combination of different antigens of the life cycle of malaria to generate more effective immune responses.

## 3. Conclusion and Future Perspectives

Recently, the world has been going through a dark moment caused by COVID-19, which has shown the state of our preparedness for future public health emergencies. It had multiple factors to make NTDs become a public health emergency of international concern, but we can never rule out human behavior and how it leads to environmental changes that directly impact our lives and health. Moreover, other factors emerged in response to the pandemic state, such as social inequalities, the increase in food insecurity, and the global economic crisis.

Besides NTDs, another constant danger is the emerging and re-emerging diseases. Some examples are the recent Ebola epidemic, a virus with high mortality rates affecting the most vulnerable African regions, and HIV, an ongoing pandemic in which there are still no effective and licensed vaccines. Lastly, the epidemics caused by the H1N1 (2009), SARS-CoV (2002), and MERS-CoV (2012–2019) viruses were already showing signs of a possible epidemic, with the risk of a pandemic.

It is imperative to invest in health research as well as to discuss public policies to prevent new epidemics and pandemics. The COVID-19 pandemic showed that the poorest and vulnerable countries were the most affected by the losses of human lives. Besides, it is necessary to discuss public policies for vaccinations to reach inaccessible areas, due to harsh natural conditions, violence, war, and other risks, as well as special groups such as prisoners, pregnant women, indigenous, the Quilombola community, and people with comorbidities (Figure 3). Another essential point at this critical moment is the increase in anti-vaccine movements and the vaccine deniers who decreased the credibility in science. We all know that science has a fantastic historical trajectory that has contributed to social evolution and technological advances. To solve this problem, society must be better educated about immunization and its biosafety (Figure 3).

In conclusion, it is worth calling attention to the fact that we, Western citizens, who enjoy a comfortable standard of living, will not continue in this apparent comfort, if we do not look at the world as a set of connected lives, and start to fight correctly against poverty and scientific apartheid. Therefore, we need to use all of the possibilities of scientific and technological development to produce vaccines capable of generating health for the general population, not just for the financially rich countries. An easy and recent example of the vaccine apartheid happened during the COVID-19 pandemic, when the ten richest countries in the world obtained more than two thirds of the vaccines against COVID-19 that were produced in the first six months after their development. The result of this was the increased viral circulation and the infection of people without vaccine protection, consequently a significant increase in the number of preventable deaths. Beyond that, the uncontrolled viral circulation provided a greater chance of viral mutations, which enabled the virus to escape the immunity caused by the vaccines. Thus, despite vaccination being one of the best tools against infectious diseases and the vaccine technologies having evolved significantly in recent decades, unfortunately human behavior makes science and what it generates, such as vaccines, an activity that serves business, rather than serving lives, as it should be.

## Figures and Tables

**Figure 1 vaccines-10-01385-f001:**
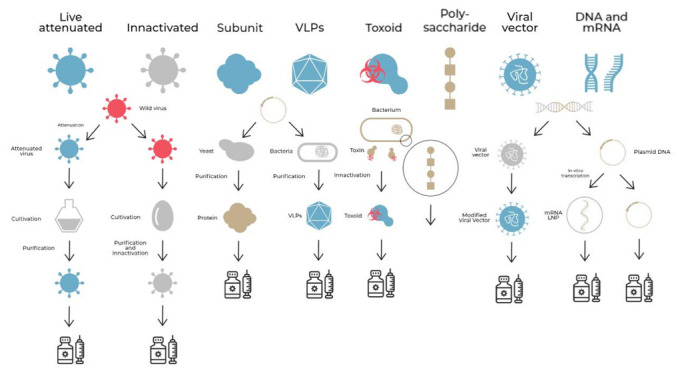
Vaccine generations and different development strategies used nowadays. VLP (Virus-Like Particle); LNP (Nanoparticle).

**Figure 2 vaccines-10-01385-f002:**
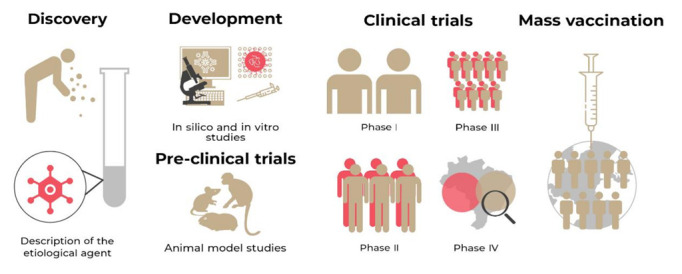
Line of development time and release of a vaccine: Discovery and description of the etiological agent; Development of the vaccine prototype; Preclinical tests in animal models; Clinical trials that are subdivided into phase I, phase II, and phase III; Approval, vaccination release, and phase IV clinical test; Mass vaccination program.

**Figure 3 vaccines-10-01385-f003:**
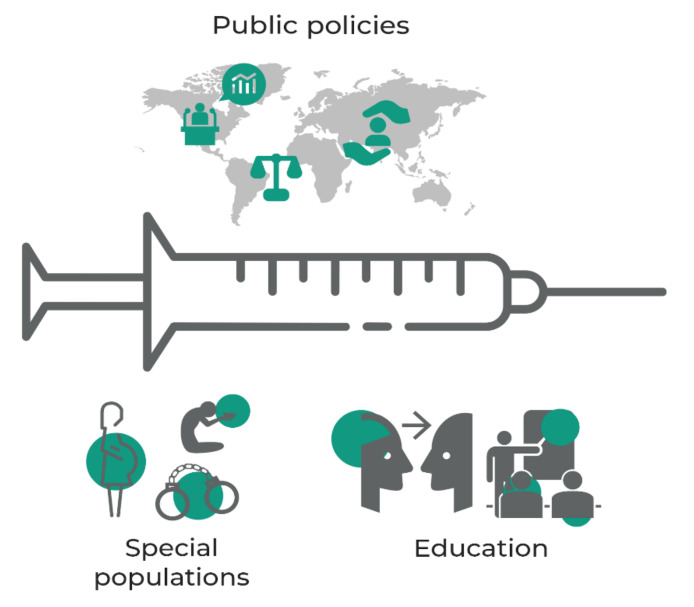
Public policies need to ensure vaccination to populations in high-risk areas where nature and social problems, such as war, violence, and poverty, make access difficult by healthcare. They also need to address special groups, including women, the poor, indigenous communities, and patients with comorbidities. However, this cannot be accomplished only by offering vaccines to the population; individuals and communities also need to understand the benefits of vaccination, which will only be accomplished by education.

**Table 1 vaccines-10-01385-t001:** Neglected Tropical Diseases (NTDs), etiological agents, and epidemiology.

Neglected Tropical Diseases (NTDs)	Etiological Agent	Epidemiology
Buruli ulcer	*Mycobacterium ulcerans* (bacteria)	1258 new cases in 2020 (1)
Chagas disease (American trypanosomiasis)	*Trypanosoma cruzi* (parasite)	6–7 million people worldwide are estimated to be infected (1)
Dengue and Chikungunya	Mosquito-borne viral infection (DENV and CHIKV)	100–400 million infections per year (1)
Dracunculiasis	*Dracunculus medinensis* (warm parasite)	27 cases in 2020 (1)
Echinococcosis	*Echinococcus granulosus* and *Echinococcus multilocularis* (parasite)	50–100.000 person-years in endemic areas (1)
Foodborne trematode infections.	*Clonorchis* sp, *Opisthorchis* sp., *Fasciola* sp. and *Paragonimus* sp. (parasites)	2 million lives lost, disability and death worldwide every year (1)
Human African trypanosomiasis (sleeping sickness)	*Trypanosoma* genus (protozoa parasite)	992 new cases in 2019 (1)
Leishmaniasis	*Leishmania* sp. (protozoa parasite)	700,000–1 million new cases per year (1)
Leprosy (Hansen’s disease)	*Mycobacterium leprae* (bacteria)	202,256 new cases in 2019 (1)
Lymphatic filariasis (Elephantiasis)	*Wuchereria bancrofti, Brugia malayi, and Brugia timori* (warm parasite)	51 million new cases in 2018 (1)
Mycetoma, chromoblastomycosis and other deep mycosis	Caused by different species of bacteria or fungi	The global charge is unknown (1)
Onchocerciasis	*Onchocerca volvulus* (worm parasite)	25 million people are infected worldwide (1)
Rabies	Rabies virus (RABV) caused by *Lyssavirus* genus	59,000 deaths worldwide each year (2)
Scabies and other ectoparasitosis	*Sarcoptes scabiei var hominis* (parasite)	200 million people worldwide suffer from scabies at any one time (1)
Schistosomiasis (Bilharzia)	*Schistosoma sp.* (warm parasite)	200,000 annual death globally (1)
Soil-transmitted helminthiases	Different species of parasitic worms (*Ascaris lumbricoides*, *Trichuris trichiura*, *Necator americanus*, and *Ancylostoma duodenale*)	More than 1.5 billion people, or 24% of the world’s population, are infected with soil-transmitted helminth infections worldwide
Snakebite envenoming	Several species of venomous snakes	1.8 to 2.7 million cases of envenomings per year (1)
Taeniasis and cysticercosis	*Taenia solium, Taenia saginata,* and *Taenia asiatica* (warm parasite)	In 2015, 2.8 million people with disability-adjusted life-years (DALYs) caused by taeniasis; 2.56–8.30 million suffered from neurocysticercosis (NCC) (1)
Trachoma	*Chlamydia trachomatis* (bacterium)	Is responsible for the blindness or visual impairment of about 1.9 million people in 44 countries
Yaws (endemic treponematoses)	*Treponema* genus	82,564 cases were reported and 153 cases were confirmed (1)

Source: (1) World Health Organization (WHO)_https://www.who.int/news-room/fact-sheets/detail_ accessed on 17 March 2022; (2) Centers for Disease Control and Prevention (CDC)_ https://www.cdc.gov/rabies/location/world/index.html_ acessed on 17 March 2022.

## Data Availability

MDPI Research Data Policies at https://www.mdpi.com/ethics.

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
