# Peer review of "Vaccines against Emerging and Neglected Infectious Diseases: An Overview"

_vaccines, 2022, doi:10.3390/vaccines10091385_

Round 1

Reviewer 1 Report

In this paper, Vuitika and colleagues summarize vaccine efforts directed towards neglected tropical diseases.  Overall this is an interesting paper and easy to read, with the exception of the final section whose focus was a little unclear.  I could see this article being a helpful portal into the primary literature.

Beyond listing the different pathogens and vaccines directed against them, the review article does not go into much depth about any one topic.  It might add value if the article went over the different vaccine platforms, their pros and cons, and why investigators have chosen certain platforms over others for particular targets.  

Author Response

We acknowledge your valuable suggestions and helpful comments. We have updated a new version of the manuscript with more information to address your valuable suggestions. We seek to delve deeper into the topics about each disease and the technologies used to develop vaccines, as well as discussing the pros and cons of each vaccine technology. In addition, we discussed the problems that can occur when it is make the wrong choice of a certain vaccine antigen, thus, we need to pay greater attention to the biological relationship between the microorganism and the host. Beyond that, we made improvement about the references.

All the changes were highlighted in yellow to be easily seeing in the manuscript.

Reviewer 2 Report

A review article “Vaccines against emerging and neglected infectious diseases: an overview” by Larissa Vuitika et al., introduce current vaccine development for 8 of neglected and emerging infectious diseases. Readers would benefit from the review on vaccine development status and strategies of each disease, current status of vaccine use to grasp overall ideas regarding field of NTD vaccine. However, the manuscript can be improved for publication. Especially, I strongly recommend to revise citation and references considering the review paper delivers the knowledge supported by references. There are many missing citations as well as errors in references. Please find detailed comments.

1. Introduction

Line 35: Match the citation and reference [2] in line 328-331.

Line 39~40: Please provide web site addresses directly lead to the information.

Line 43~44: Sentence fragment?

Line 54 and 61: Please check the citation; [5] was cited in line 54 and the next citation is [29].

Line 64~65: The sentence is ambiguous.

Line 71: Vaccination might be one of the effective measures to control infectious diseases.

Line 75: “immune protection” can be “protective immune response”.

Line 76~77: According to line 32~35, NTD prevalence is restricted to tropical regions (countries) which seems rather contradictory to “importance of accessibility of NTD vaccines”.

Line 77~81: Please check grammar.

Line 83~84: Some of abbreviations, HVsAg and Al in the legend, are not used in Figure 1.

Line 86~89: Figure numbers, (1)~(6), are not shown in Figure.

Line 128: people “,who are” over nine years of age and is seropositive for dengue.

Line 128~130: Detailed explanation is required why “peopled aged two to five who develop severe disease when expose to the virus for the second time” is the limitation for the vaccination.

Line 134, 136: Is “non-service protein” common term? Otherwise replace to “nonstructural protein”. Replace “NSI” to “NS1”.

2. An overview of NTDs and vaccination

Line 95: Recommend to reconsider “vaccination” since contents are mainly address the current status of vaccine development.

Line 123, 149, 170, 190: numbering error of section title

Line 124: due to “the” four antigenically

Line 141: What is “DMV-2”?

Line 155: Please check citation format.

Line 165: “to” prevent CHIKV “infection”. Omit “already”.

Line 168~169: “Other three vaccines, using the attenuated CHIKV (add strain name if available), the measles virus vector, and the chimpanzee adenovirus vector, are in the phase (add stage information if possible) of clinical studies”.

Line 178~179: correct the tense of “show” and add citation.

Line 194~195: Replace “37,7 million” to “37.7 million”, “which” to “those”, and remove “(25.4 million)”.

Line 214~216: Please rephrase the sentence.

Line 232~234: Please rephrase the sentence.

Line 241: “where” to “of which”

Line 234: “from human-human” to “human to human”

Line 241~243: Sand fly is the transmission vector between animal and human and leishmania replicates in dogs and human.

Line 257: RTS,S/AS01 was approved in 2015 by EMA and endorsed by WHO in Oct, 2021 for broad use.

Line 263, 269, 275: Reference [35] and [73] are the same paper.

Line 277~278: Please rephrase the sentence.

Line 280~281: “blocking” is redundant.

3. Conclusion and future perspectives

Line 282: Correct section number from “6” to “3”.

Line 283: “the need for preparation” to “the preparedness”

Line 285~286: Please clarify how the globalization led to vulnerability to new harmful microorganisms.

Line 286~291: These inferences need supportive citations. Besides, I do not understand what authors intend to claim in line 288~291.

Line 298: “try preventing” to “prevent”

Line 299~300: Add citation to support the sentences.

Line 304: Contents and Figure 1 do not match.

Line 311~315: Figure legend is redundant with main body.

Line 298~309: I am not sure last paragraph well sum up the whole story about NTD vaccines.   

References

There are many uncited references and if so, those should be deleted.

Line 328~331: There are two references assigned in [2].

Author Response

We acknowledge your valuable suggestions and helpful comments. We have updated a new version of the manuscript with more information to address your valuable suggestions. All the changes were highlighted in yellow to be easily seeing in the manuscript.

It is important to state that all your comments have been considered as very valuable and we have tried to update the article, in a way that the final version of the manuscript will be better presented to the public. For example, besides accept and change the manuscript based on your suggestions, we delve deeper into each topics discussed, such as disease and the technologies used to develop vaccines, as well as discussing the pros and cons of each vaccine technology. In addition, we discussed the problems that can occur when it is make the wrong choice of a certain vaccine antigen, thus, we need to pay greater attention to the biological relationship between the microorganism and the host. Beyond that, we made improvement about the references, as requested.